# Incidence of Differentiation Syndrome Associated with Treatment Regimens in Acute Myeloid Leukemia: A Systematic Review of the Literature

**DOI:** 10.3390/jcm9103342

**Published:** 2020-10-18

**Authors:** Lucia Gasparovic, Stefan Weiler, Lukas Higi, Andrea M. Burden

**Affiliations:** 1Institute of Pharmaceutical Sciences, Department of Chemistry and Applied Biosciences, ETH Zurich, 8093 Zurich, Switzerland; glucia@student.ethz.ch (L.G.); Stefan.weiler@toxinfo.ch (S.W.); lhigi@student.ethz.ch (L.H.); 2National Poisons Information Centre, Tox Info Suisse, Associated Institute of the University of Zurich, 8032 Zurich, Switzerland

**Keywords:** differentiation syndrome, retinoic acid syndrome, differentiating agents, acute myeloid leukemia, non-M3 acute myeloid leukemia, acute promyelocytic leukemia, systematic review

## Abstract

Differentiation syndrome (DS) is a potentially fatal adverse drug reaction caused by the so-called differentiating agents such as all-trans retinoic acid (ATRA) and arsenic trioxide (ATO), used for remission induction in the treatment of the M3 subtype of acute myeloid leukemia (AML), acute promyelocytic leukemia (APL). However, recent DS reports in trials of isocitrate dehydrogenase (IDH)-inhibitor drugs in patients with IDH-mutated AML have raised concerns. Given the limited knowledge of the incidence of DS with differentiating agents, we conducted a systematic literature review of clinical trials with reports of DS to provide a comprehensive overview of the medications associated with DS. In particular, we focused on the incidence of DS reported among the IDH-inhibitors, compared to existing ATRA and ATO therapies. We identified 44 published articles, encompassing 39 clinical trials, including 6949 patients. Overall, the cumulative incidence of DS across all treatment regimens was 17.7%. Incidence of DS was notably lower in trials with IDH-inhibitors (10.4%) compared to other regimens, including ATRA and/or ATO (15.4–20.6%). Compared to other therapies, the median time to onset was four times longer with IDH-inhibitors (48 vs. 11 days). Treating oncologists should be mindful of this potentially fatal adverse drug reaction, as we expect the current trials represent an underestimation of the actual incidence.

## 1. Introduction

Differentiation syndrome (DS) is a potentially fatal adverse drug reaction caused by the so-called differentiating agents, such as all-trans retinoic acid (ATRA) and arsenic trioxide (ATO) drugs, which control cellular differentiation and proliferation. The primary use of ATRA and ATO is for remission induction in treating the M3 subtype of acute myeloid leukemia (AML), acute promyelocytic leukemia (APL) [1,2]. APL accounts for approximately 5 to 20% of all AML cases, and age at diagnosis is significantly lower than in other types of AML, laying between 20 and 50 years [3]. If untreated, the median survival of APL is one week [4]. However, if treated early with the recommended induction therapy, consisting of ATRA and an anthracycline (idarubicin or daunorubicin) or ATRA and ATO, APL is a highly curable disease [5]. 

Despite the dramatic impact of ATRA and ATO in the treatment of APL, the causes of treatment failures during induction therapy include fatalities resulting from hemorrhage and infection as general complications of chemotherapeutic therapy. Moreover, the adverse drug reaction DS is a serious and specific complication that can result from ATRA treatment of APL. While approximately half of early APL deaths are attributable to hemorrhage due to coagulopathy, DS remains one of the most frequently cited causes of death [6]. Up to a quarter of APL patients will experience DS during their treatment, yet evidence on the pathophysiology, epidemiology, predictive factors, and prevention is still scarce [1,7,8].

DS was first described by Frankel et al. in 1992 as a distinct symptom complex characterized primarily by fever and dyspnea in the absence of infection, occurring during the treatment of patients treated with ATRA for APL [9]. Other symptoms were peripheral oedema, rapid weight gain, unexplained episodic hypotension, renal insufficiency, and hyperbilirubinemia. Frankel et al. noted that the syndrome only correlated with leukocytosis in some but not all patients and that the administration of intravenous dexamethasone brought rapid relief in most patients. While the authors named the symptom complex “retinoic acid syndrome” after the causative drug [9], “differentiation syndrome” is more commonly used in recent years [10,11].

Although DS is rare within the overall population, it occurs in up to one-quarter of patients treated with the differentiating agents, ATRA and ATO, making it a highly significant adverse drug reaction in these patients [7]. More recently, two clinical trials of isocitrate dehydrogenase (IDH)-inhibitor drugs for the treatment of relapsed/refractory IDH2-mutated (non-M3) AML unexpectedly observed DS as an adverse event, thereby identifying the IDH-inhibitors as potential differentiating agents [12,13,14,15]. The IDH-inhibitors enasidenib (IDHIFA^®^, Celgene) and ivosidenib (TIBSOVO^®^, Agios Pharmaceuticals) were approved by the US Food and Drug Administration (FDA) in 2017 and 2019, respectively. Due to the unexpected findings, the FDA included a boxed warning for the potential of DS with enasidenib in 2018 [16,17,18,19]. Additionally, the marketing authorization application of enasidenib submitted to the European Medicines Agency (EMA) was withdrawn by Celgene in December of 2019 due to a lack of a positive benefit/risk ratio in the proposed indication [20].

As there is limited knowledge of DS incidence, particularly among the non-M3 AML, we aimed to complete a systematic review on the incidence of DS with all treatment regimens in AML patients. In particular, we focused on the new IDH-inhibitor drugs compared to existing differentiating therapies based on ATRA and ATO to contextualize the occurrence of DS in these new agents.

## 2. Methods

### 2.1. Literature Search and Article Selection

We identified all studies published from 1 January 1992 through to 25 June 2019. The start date corresponds to the first report of DS published by Frankel et al. in 1992 [9]. We searched the PubMed database using the search string *(“differentiation syndrome”[tw]) OR (different syndrom*[tw]) OR (“retinoic acid syndrome”[tw]) OR (retin acid syndrom*[tw]) AND (Humans[Mesh])*. Articles were excluded if they were not published in English if they did not contain any information on the administered drug, and/or if they contained no, or insufficient, information on the occurrence of DS. We further excluded abstracts, commentaries, letters to the editor, and review articles. While we took all efforts to retrieve the full-text of all articles selected during the screening of title and abstract, we excluded articles that were not available for full-text review.

Screening of the articles was repeated independently by two authors (LG, LH), with disagreement resolved through discussion or consultation with a third author (AMB). While we identified all relevant publications, only clinical trials were eligible for detailed review. Furthermore, one author (LG) conducted a hand search of the reference lists among the selected articles retrieved from PubMed. Additionally, all included articles were cross-referenced with a retraction database to ensure none of the articles had retraction notices [21].

### 2.2. Data Extraction

The authors used a standardized data extraction format. The data extraction was performed per trial, combining information from multiple publications in one trial record, if necessary. Extracted information included patient demographics and information on the study treatment (administered drugs, doses, route, frequency, and duration). Furthermore, we extracted the number of patients who had experienced DS and the number of patients who died. If available, we extracted the time to onset of DS, as well as the treatment of DS. Clinical trial treatment arms were grouped according to their treatment regimen into “ATRA only,” “ATO only,” “ATRA + ATO,” “ATRA + cytotoxic chemotherapy” (“ATRA + CT”), “IDH-inhibitors,” and “retinoic acid derivatives.” We categorized regimens that did not fit the above categories as “other.”

### 2.3. Risk of Bias Assessment

We assessed all included clinical trials for the risk of bias using a modified version 2 of the Cochrane Risk of Bias Tool for randomized trials (RoB 2) [22]. The risk of bias assessment was conducted by the primary author (LG) for the reported primary outcome of the respective clinical trial and aimed to estimate the risk of bias on adhering to the intervention. We used information from the published journal article(s). However, when available, we used additional sources, such as clinical trial databases. We then assessed the level of bias that was likely (high) or unlikely (low) to impact the study results. The robvis package for R (3.5.2) generated the summary figures of the risk of bias assessment.

### 2.4. Statistical Analysis

We calculated the incidence of DS as a percentage of the whole study population. Additionally, we calculated the weighted mean cumulative incidence of DS in percent and the corresponding 95% confidence intervals (95% CI) within each treatment regimen. For the calculations of the cumulative incidence of DS, we considered only patients who had received a differentiating agent as part of their treatment regimen. Patients in treatment arms, not including a differentiating agent, were excluded from the calculation of the cumulative incidence of DS. We stratified results by the individual treatment arms (ATRA + ATO, ATRA + CT, ATRA, ATO, IDH-inhibitors, retinoic acid derivatives, other).

## 3. Results

We identified a total of 838 articles from PubMed; after removing duplicates and non-English studies, 591 titles and abstracts were screened, resulting in the exclusion of 434 articles (Figure 1). In addition to the remaining 157 full-text articles, we further identified 57 articles [12,23,24,25,26,27,28,29,30,31,32,33,34,35,36,37,38,39,40,41,42,43,44,45,46,47,48,49,50,51,52,53,54,55,56,57,58,59,60,61,62,63,64,65,66,67,68,69,70,71,72,73,74,75,76,77,78] from the hand-search of the reference lists, resulting in 214 full-text articles. Of these, we excluded 80. Of the 134 eligible publications, 44 were clinical trial reports and included in the final analysis [1,12,13,27,29,31,34,40,43,44,48,49,51,52,53,54,55,56,57,64,68,70,72,74,75,76,77,79,80,81,82,83,84,85,86,87,88,89,90,91,92,93,94,95]. The remaining 90 were case reports (*n* = 67) or observational study reports (*n* = 23).

Characteristics of the 44 included publications, reporting on 39 unique clinical trials, are summarized in Table 1 and Appendix A. In the 39 clinical trials included in the systematic review, we identified a total of 6949 patients. Sex distribution was almost equal with 51.5% male (*n* = 3577) and 47.9% female (*n* = 3327) patients (0.6% unknown, *n* = 45). The median ages ranged from 8 to 70 years in all 39 clinical trials. The three trials conducted in patients with non-M3 AML had reported median ages of 68, 70, and 69, respectively [12,13,93]. The highest median age in the clinical trials, including only APL patients, was 65 [84].

Of all 6728 patients in clinical trials treated with a differentiating agent, a mean of 17.7% (95% CI 14.9–20.4) experienced DS. Of all patients in clinical trials treated with ATRA and ATO concomitantly (*n* = 959, 14.3%), a weighted mean of 15.4% (95% CI 10.2–20.7) experienced DS. If treated with ATRA and concomitant cytotoxic chemotherapy (ATRA + CT) (*n* = 3536, 52.6%), DS occurred in a mean of 19.6% (95% CI 14.2–25.0) of patients. Treatment with single-agent ATRA (*n* = 616, 9.2%) or ATO (*n* = 384, 5.7%) resulted in cumulative DS incidences of 19.5% (95% CI 13.7–25.3) and 20.6% (95% CI 12.9–28.2), respectively. In the clinical trials of the IDH-inhibitors ivosidenib and enasidenib, comprising a total of 497 patients (7.4%), DS occurred in a mean of 10.4% (95% CI 9.3–11.5) of patients. Trial treatment with the retinoic acid derivatives bexarotene and tamibarotene led to DS in a mean of 5.9% (95% CI 3.6–8.2) of the study population (*n* = 51).

Of the 39 clinical trials, three (7.7%) had a low risk of bias, four had some concerns (10.2%), and 32(82.1%) had a high risk of bias. Figure 2 illustrates the risk of bias by study and treatment group. The primary driver of the high risk of bias was the randomization process domain (Figure 3). It is to be noted that most of the included clinical trials were not randomized but instead used a risk- or age-based approach for assignment of the intervention or did not have a control arm, for example, the included phase 1 and 2 trials of ivosidenib and enasidenib. Recalculating the mean weighted incidence of DS from the clinical trials with a low risk of bias [40,51,52,53] or some concerns [31,34,43,68,82] only, 17.4% of patients treated with ATRA + ATO, 29.3% of patients treated with ATRA + CT, and 25.6% of patients treated with single-agent ATRA experienced DS (results not shown).

## 4. Discussion

In this review, we examined the incidence of DS with all differentiating agents, enabling an assessment of the incidence with the IDH-inhibitors compared to older existing therapies traditionally used in APL treatment. Overall, the cumulative incidence of DS in all therapies was 17.7%. DS’s incidence was notably lower in trials with IDH-inhibitors (10.4%) and retinoic acid derivatives (5.9%) than regimens based on the old differentiating agents—namely ATRA and ATO. The overall mortality rate among those with DS was relatively low, with no deaths reported among the IDH-inhibitors. The reported median time to onset was substantially longer among the IDH-inhibitor enasidenib (48 days) than the other existing therapies (11 days) (Appendix A).

In our review, we identified that most studies and treatment arms were for the combination use of ATRA and chemotherapy. This was an expected finding, as it was the recommended first-line treatment regimen for all APL patients before the approval of ATO in combination with ATRA for the first-line treatment of low- and intermediate-risk APL in 2018 [2,96]. Dividing the patients into their respective treatment groups, we observed no difference in DS incidence between the groups treated with regimens based on the old differentiating agents (ATRA and ATO), thereby indicating that no one of the ATRA- and ATO-based treatment regimens seem to pose an increased risk for DS development compared to each other. We also did not observe substantial differences in the proportion of deaths or time to DS onset among the ATRA and ATO treatment regimens.

In the patients treated with the new IDH-inhibitors ivosidenib and enasidenib and the patients treated with the retinoic acid derivatives, tamibarotene, and bexarotene, the incidence of DS was below the overall cumulative incidence. We note that the IDH-inhibitors, contrary to the ATRA- and ATO-based regimens, were used to treat non-M3 AML, as was the retinoic acid derivative bexarotene. Therefore, it is possible that non-M3 AML patients are less prone to developing DS during their remission induction. What might further contribute to a lower DS incidence in IDH-inhibitors and retinoic acid derivatives is that three of the four clinical trials were phase 1 and using dose escalation, meaning that part of the patients received very low doses of the differentiating drugs. Since our estimates of the incidence of DS in the treatment with the IDH-inhibitors, ivosidenib and enasidenib, is based on the published phase 1/2 clinical trials, we may be underestimating the real-world incidence and mortality of DS associated with IDH-inhibitors [13,14,91]. 

Additionally, the delayed time to onset observed in the IDH-inhibitors indicates that either DS develops later in these patients, or there was a delay in clinical detection. A recent systematic analysis by the US FDA conducted an algorithmic analysis to improve the capture of DS cases in patients using IDH-inhibitors as many of the signs and symptoms of DS may not be initially recognized [14]. The authors concluded that DS was likely common with the IDH-inhibitors, and increased awareness of the early signs and symptoms is needed to decrease the potential for severe or fatal complications. Lastly, due to the lack of diagnostic markers, the diagnosis of DS in ATRA and ATO treatment relies on several unspecific symptoms. Whether these diagnostic criteria can be applied to detect DS in non-M3 AML patients reliably, or whether the application of the same criteria might lead to further underdiagnosis, is unclear [1]. Moreover, we note that the complete remission rates were substantially higher in the ATRA/ATO patients than those treated with IDH-inhibitors. Thus, without clinical awareness and standardized diagnostic criteria of DS with IDH-inhibitors, it is possible that the symptoms of DS are incorrectly categorized as disease progression and subsequently under-recorded. Thus, without clinical awareness and standardized diagnostic criteria of DS with IDH-inhibitors, it is possible that the symptoms of DS are incorrectly categorized as disease progression and subsequently under-recorded. Given this, we believe the FDA boxed warning [18,19] is an appropriate measure to alert health professionals to this potentially fatal ADR to ensure time and appropriate treatment [18,19]. We also note that the market authorization application submitted to the EMA for enasidenib was withdrawn due to a limited benefit/risk ratio [20]. Thus, it is evident that large clinical trials, pharmacovigilance, and post-market observational studies of ivosidenib and enasidenib are necessary to characterize the risk of DS.

Our systematic literature review identified that the first reporting of DS in non-M3 AML was in 2008 in patients treated with bexarotene [93,97], an approved therapy for treating cutaneous T-cell lymphomas resistant to at least one prior systemic therapy [98]. In 2008, Tsai et al. reported on a phase 1 study of bexarotene conducted in patients with non-M3 AML, where two of the 27 patients in the trial experienced DS [93]. We did not identify any literature reports of bexarotene provoking DS in its approved indication. From the preclinical studies, the mechanism of action of both enasidenib and ivosidenib induces differentiation in cancer cells providing pathophysiologic plausibility to the observed occurrence of DS outside of APL [13,91]. Moreover, we saw that DS was associated with two additional drugs trialed to treat APL, realgar-indigo naturalis formulation (RIF), and tamibarotene. In two of the ATRA trials, we identified two studies, including RIF, a traditional Chinese medicine with the main active ingredients tetraarsenic tetrasulfide (As_4_S_4_), indirubin, and tanshinone IIA [40,53]. In both studies from China, RIF was an oral replacement for ATO [99]. Tamibarotene is a derivative of retinoic acid and approved in Japan to treat relapsed and refractory APL [100]. Currently, a phase 2 trial of tamibarotene for the treatment of AML and myelodysplastic syndrome is ongoing in the USA (NCT02807558) [101] and was included in our review as a single-agent drug in a phase 1 study as a second-line treatment [88]. A multicenter phase 2 study on tamibarotene in adults with relapsed or refractory APL after treatment with ATRA and ATO (NCT00520208) found an overall response rate of 64%. However, the median overall survival was 9.5 months, and 7 of 9 patients relapsed after a median of 4.6 months [102].

Due to the inclusion of many non-randomized clinical trials, most of the clinical trials in this systematic review had a high risk of bias. However, we believe the high risk of bias noted does not necessarily disqualify these studies or their results but rather presents a limitation to the assessment tool. However, we note only marginal changes in the weighted mean incidence when including only those trials with a low risk of bias [40,51,52,53] or some concerns [31,34,43,68,82]. The most considerable change was among patients treated with ATRA+CT (19.6% vs. 29.3%), which is likely due to the inclusion of only three studies with DS incidence ranging from 5.6% to 36.8%.

Our systematic review has notable limitations. We performed a single-engine search in PubMed, which primarily indexes medical literature and comprises more than 30 million citations [103]. As such, we may have missed some articles that are not PubMed indexed. However, within PubMed, we chose inclusive search terms with no exclusion based on disease type. Additionally, we conducted a thorough citation search of all relevant articles, which identified 25 additional articles to minimize this risk. As we note that this may limit the reproducibility of the results, we have provided the citations of the articles identified by hand search to improve transparency. Given the small number of studies with appropriate randomization, the clinical trials were subject to a high risk of bias. Although we did not identify substantial differences between the findings of those trials with a low or moderate risk of bias, as compared to the high risk of bias, we believe future analyses of large randomized trials are needed, particularly for the new IDH-inhibitors.

Moreover, while the outcome of DS incidence did not change when assessing those with a low or moderate risk of bias, we expect that clinical trial reports are subject to misclassification bias, rooted in the lack of universally applied diagnostic criteria for DS and the unspecific nature of its symptoms, particularly for the non-M3 AML patients treated with IDH-inhibitors [7,104]. Another limitation is that we focused our review on the occurrence of DS and, therefore, did not summarize information on disease-free survival or relapse rate. We note that there is no demonstrated benefit of ATRA combination with conventional chemotherapy, and consolidation in randomized studies is needed. However, clinical trials have shown a statistically significant reduction in relapse rates and longer disease-free survival [30,70].

Finally, while we acknowledge the limitation of comparing treatment regimens for M3 and non-M3 acute leukemia, we believe a review comparing different treatment regimens causing DS improves the understanding of this potentially fatal adverse drug reaction and highlights the significance in the newly developed IDH-inhibitors. Therefore, we hope to raise awareness of DS among treating clinicians and encourage researchers to conduct observational studies to provide real-world clinical evidence to further elucidate the development of DS in both M3 and non-M3 AML. Moreover, future research undertaking clinical development of potential differentiating agents should consider and assess the potential for DS. We believe this is necessary considering that differentiating therapy is a promising and evolving therapeutic field [17], yet the unspecific nature of DS symptoms makes it difficult to diagnose if occurring unexpectedly in non-traditional patient groups.

## 5. Conclusions

This research aimed to contextualize the unexpected occurrence of DS in the clinical trials, with a particular focus on the comparison of IDH-inhibitors (ivosidenib and enasidenib) to existing ATRA and ATO therapies. While we found no difference between the ATRA- and ATO-based regimens used in APL treatment, DS occurred less frequently during treatment with IDH-inhibitors in non-M3 AML. However, this may be due to underreporting and lack of existing diagnostic criteria in this patient group. In light of the unexpected occurrence of DS in non-M3 AML, and considering that differentiation therapy is a promising research area expected to contribute to future cancer treatments, we consider it crucial to raise awareness of DS among treating clinicians and researchers to close this knowledge gap. Due to the paucity of data and rarity of DS in IDH-inhibitors, we believe the warnings from competent authorities are justified, and treating oncologists should be mindful of the potential of DS in non-M3 AML patients. However, further large randomized trials or studies using real-world data are required to provide further insights.

## Figures and Tables

**Figure 1 jcm-09-03342-f001:**
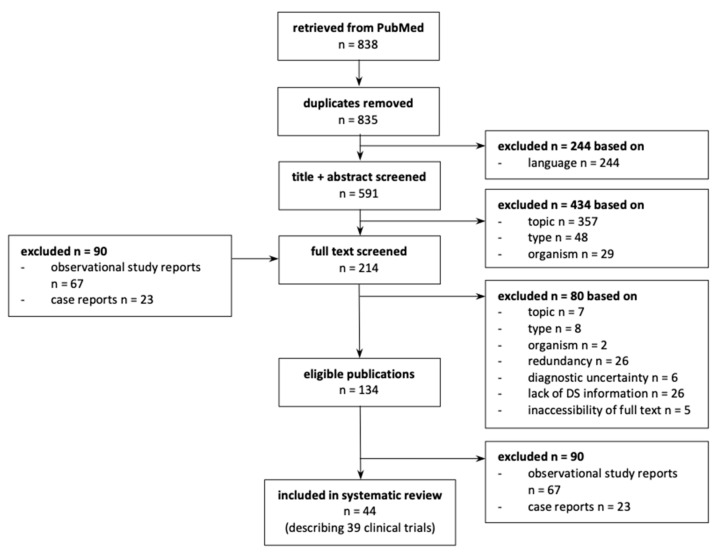
Article selection flow diagram. Redundant described a study when the original report was present, and the article did not add any further information. Of five articles, a full-text version was not available to our institution.

**Figure 2 jcm-09-03342-f002:**
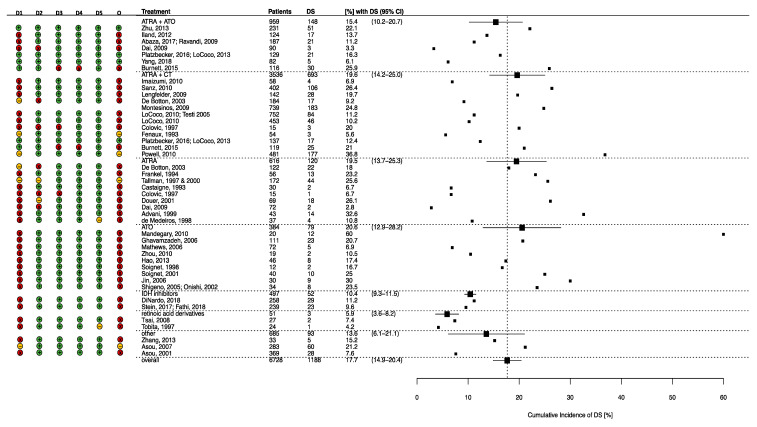
Cumulative incidence of Differentiation Syndrome in the clinical trials grouped by treatment regimen arm. Abbreviations: D = Dimension, D1–D5: Dimensions derived from Risk of Bias Tool (D1: bias arising from the randomization process; D2: Bias due to deviations from intended interventions; D3: Bias due to missing outcome data; D4: Bias in the measurement of outcome; D5: Bias in the selection of cases). O: Overall risk of bias assessment. DS: differentiation syndrome, CI: confidence interval, ATRA: all-trans retinoic acid, ATO: arsenic trioxide, CT: chemotherapy, IDH: isocitrate dehydrogenase. Notes: Grouped by treatment arm, not individual trials. Green indicates a low risk of bias; Yellow indicates some concerns; Red indicates a high risk of bias. The risk of bias was generated using the risk of bias visualization tool (https://www.riskofbias.info/welcome/robvis-visualization-tool).

**Figure 3 jcm-09-03342-f003:**
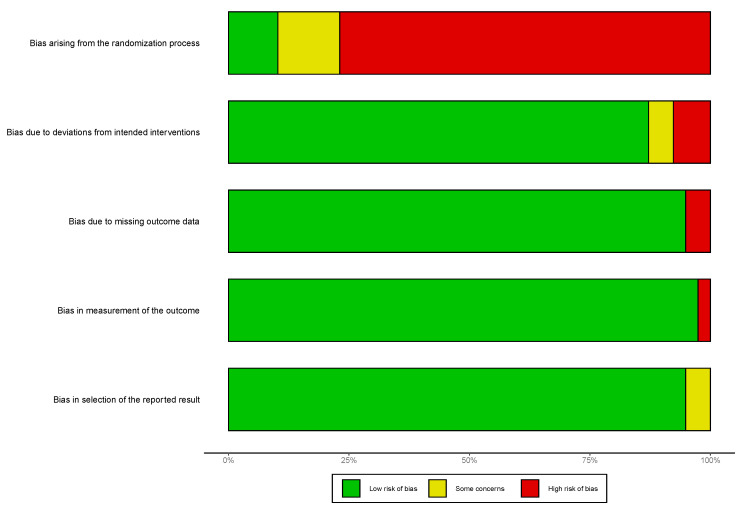
Risk of bias summary for all included studies (*n* = 39). Note: Figure generated using the Risk of Bias visualization tool available at: https://www.riskofbias.info/welcome/robvis-visualization-tool.

**Table 1 jcm-09-03342-t001:** Overview of included clinical trials (*n* = 39) and patient characteristics, treatment regimens, and outcomes.

Clinical Trials	Patients				Treatment					% DS	% Deaths	T.T.O. [d]	(Range)
Reference	Total	Age	(Range)	% Male	Allocation	Primary Treatment							
Zhu, 2013 [53]	231	37	(15–60)	54.5						22.1	0		
	117	39	(15–60)	55.6	randomized	ATRA 25 mg/m^2^	ATO 0.16 mg/kg			24.8	0		
	114	33	(15–60)	53.5	randomized	ATRA 25 mg/m^2^	RIF 60 mg/kg			19.3	0		
Iland, 2012 [87]	124	44	(3–78)	50.0						13.7	0		
	108				≤60 y.	ATRA 45 mg/m^2^/d tid	ATO 0.15 mg/kg/d qd d9-36	idarubicin 12 mg/m^2^/d d2,4,6,8					
	9				61–70 y.	ATRA 45 mg/m^2^/d tid	ATO 0.15 mg/kg/d qd d9-36	idarubicin 9 mg/m^2^/d d2,4,6,8					
	7				>70 y.	ATRA 45 mg/m^2^/d tid	ATO 0.15 mg/kg/d qd d9-36	idarubicin 6 mg/m^2^/d d2,4,6,8					
Abaza, 2017 [49]; Ravandi, 2009 [55]	187	50	(14–84)	51.9						11.2	0		
					WBC < 10 × 10^9/l^ old	ATRA 45 mg/m^2^/d bid	ATO 0.15 mg/kg/d qd from d10						
					WBC ≥ 10 × 10^9/^l old	ATRA 45 mg/m^2^/d bid	ATO 0.15 mg/kg/d qd from d10	Gemtuzumab ozogamicin 9 mg/m^2^/d qd d1					
					WBC < 10 × 10^9/l^ new	ATRA 45 mg/m^2^/d bid	ATO 0.15 mg/kg/d qd from d1						
					WBC ≥ 10 × 10^9/l^ new	ATRA 45 mg/m^2^/d bid	ATO 0.15 mg/kg/d qd from d1	Gemtuzumab ozogamicin 9 mg/m^2^/d qd d1					
Dai, 2009 [54]	162			57.4						3.1	0		
	72	34	(14–69)	54.2		ATRA 45 mg/m^2^/d				2.8			
	90	32	(14–67)	60.0		ATRA 45 mg/m^2^/d	ATO 10 mg/d d1-28			3.3			
Platzbecker, 2016 [51]; LoCoco, 2013 [52]	266			48.9						14.3	0.8		
	129	46.6	(19–70)	46.5	randomized	ATRA 45 mg/m^2^/d bid	ATO 0.15 mg/kg/d qd			16.3	0		
	137	46.6	(18–70)	51.1	randomized	ATRA 45 mg/m^2^/d bid	idarubicin 12 mg/m^2^/d qd d2,4,6,8			12.4	1.5		
Yang, 2018 [40]	82	9.4	(1–16)	62.2						6.1			
	42	7.8	(1–13)	69.0	randomized	ATRA 25 mg/m^2^/d	mitoxantrone 10 or 7 mg/m^2^/d d3 or d2-4	ATO 0.16 mg/kg/d qd from d5/6		9.5			
	40	9.9	(2–16)	55.0	randomized	ATRA 25 mg/m^2^/d	mitoxantrone 10 or 7 mg/m^2^/d d3 or d2-4	RIF1 35 mg/kg/d tid from d5/6		2.5			
Burnett, 2015 [77]	235	47	(16–77)	51.1						23.4	0		
	119	47	(16–77)	50.4	randomized	ATRA 45 mg/m^2^/d bid	idarubicin 12 mg/m^2^/d qd d2,4,6,8			21.0			
	116	47	(16–75)	51.7	randomized	ATRA 45 mg/m^2^/d bid	ATO 0.3 mg/kg/d qd d1-5 & wk2-8			25.9			
Imaizumi, 2010 [92]	58	11	(1–16)	53.4		ATRA 45 mg/m^2^/d	daunorubicin 45 mg/m^2^/d d6-8	cytarabine 200 mg/m^2^/d d6-12		7.3	0		
Sanz, 2010 [44]	402	42	(3–83)	52.0						28.5	1.1		
					20–70 y.	ATRA 45 mg/m^2^/d bid	idarubicin 12 mg/m^2^/d qd d2,4,6,8						
	22				>70 y.	ATRA 45 mg/m^2^/d bid	idarubicin 12 mg/m^2^/d qd d2,4,6						
					<20 y.	ATRA 25 mg/m^2^/d bid	idarubicin 12 mg/m^2^/d qd d2,4,6,8						
Lengfelder, 2009 [74]	142	40	(16–60)	41.5	1st induction cycle	ATRA 45 mg/m^2^/d	6-thioguanine 100 mg/m^2^ bid d3-9	cytarabine 100 mg/m^2^ d1-2, bid d3-8	daunorubicin 60 mg/m^2^ d3-5	21.1	0.8		
	131				2nd induction cycle	cytarabine 3 g/m^2^ bid d21-23	mitoxantrone 10 mg/m^2^ d23-25						
De Botton, 2003 [57]	306			44.1						12.7	1.3		
	122	45.5	(35–54)	45.9	randomized	ATRA 45 mg/m^2^/d				18.0	2.5	10	
	184	45	(34–55)	42.9	randomized	ATRA 45 mg/m^2^/d	daunorubicin 60 mg/kg/d d3-5	cytarabine 200 mg/m^2^/d d3-9		9.2	0.5	10.5	
Montesinos, 2009 [1]	739	40	(2–83)	50.6						24.8	1.4	12	(0–46)
					20–70 y.	ATRA 45 mg/m^2^/d bid	idarubicin 12 mg/m^2^/d qd d2,4,6,8						
					>70 y.	ATRA 45 mg/m^2^/d bid	idarubicin 12 mg/m^2^/d qd d2,4,6						
					<20 y.	ATRA 25 mg/m^2^/d bid	idarubicin 12 mg/m^2^/d qd d2,4,6,8						
LoCoco, 2010 [70]; Testi 2005 [56]	752			53.7						11.3	0.1		
	642	38.2	(18–61)	54.4	≥18 y.	ATRA 45 mg/m^2^/d	idarubicin 12 mg/m^2^/d qd d2,4,6,8			12.9	0.2		
	110	11.6	(1–18)	50.0	<18 y.	ATRA 25 mg/m^2^/d	idarubicin 12 mg/m^2^/d qd d2,4,6,8			1.8	0	7.5	(4–11)
LoCoco, 2010 [70]	453	40.9	(18–61)	50.6		ATRA 45 mg/m^2^/d	idarubicin 12 mg/m^2^/d qd d2,4,6,8			10.3	0.2		
Colovic, 1997 [90]	30			40.0						13.3	13.3		
	15	40	(16–65)	26.7	WBC < 5 × 10^9/l^	ATRA 45 mg/m^2^/d bid				6.7	6.7		
	15	40	(18–60)	53.3	WBC > 5 × 10^9/l^	ATRA 45 mg/m^2^/d bid	daunorubicin 50 mg/m^2^/d 3d	cytarabine 200 mg/m^2^/d 7d		20.0	20		
Fenaux, 1993 [43]	101	40	(6–67)	52.5									
	54	41.5	(6–63)	55.6	randomized	ATRA 45 mg/m^2^/d	daunorubicin 60 mg/m^2^/d 3d	cytarabine 200 mg/m^2^/d 7d		5.6	0	20	(14–24)
	47	40	(17–67)	48.9	randomized		daunorubicin 60 mg/m^2^/d 2 × 3d	cytarabine 200 mg/m^2^/d 2 × 7d					
Powell, 2010 [31]	481		(15–79)	51.4		ATRA 45 mg/m^2^/d bid	daunorubicin 50 mg/m^2^/d qd d3-6	cytarabine 200 mg/m^2^/d d3-9		36.8			
Frankel, 1994 [81]	56		(9–75)	44.6		ATRA 45 mg/m^2^/d bid				23.2	8.9		
Tallman, 1997 [68]; Tallman, 2000 [82]	346			51.7									
	172	37	(1–81)	47.7	randomized	ATRA 45 mg/m^2^/d bid				26.3	1.2	11	(2–47)
	174	38	(1–74)	55.7	randomized	daunorubicin 45 mg/m^2^/d qd d1-36	cytarabine 100 mg/m^2^/d d1-7						
Castaigne, 1993 [89]	30	56	(10–81)	43.3		ATRA 25 mg/m^2^/d bid				6.7	6.7		
Douer, 2001 [77]	69	44	(5–82)	58.0		liposomal ATRA 90 mg/m^2^qad				26.1	1.4		
Advani, 1999 [94]	43		(7–60)	65.1		ATRA 45 mg/m^2^/d				32.6	14.0	10	(4–26)
de Medeiros, 1998 [79]	37	17.5	(9–69)	45.9		ATRA 45 mg/m^2^/d				10.8	0		
Mandegary, 2010 [86]	20	31	(15–62)	35.0		ATO 15 mg/kg/d qd				60.0	5.0		
Ghavamzadeh, 2006 [72]	111	27	(6–79)	45.9		ATO 15 mg/kg/d qd				20.7	7.2		
Mathews, 2006 [64]	72	28	(3–75)	52.8						6.9	0	13.2 *	(6–21)
					adults	ATO 10 mg/d	hydroxyurea 0–4 g/d						
					pediatric patients	ATO 0.15 mg/kg/d	hydroxyurea 0–30 mg/kg/d qd-qid						
Zhou, 2010 [27]	19	10	(4–15)	57.9						10.5	0		
	5				4–6 y.	ATO 0.2 mg/kg/d qd							
	14				>6 y.	ATO 0.16 mg/kg/d qd							
Hao, 2013 [95]	46	8	(mean)	76.1		ATO 0.17–0.33 mg/kg/d qd				17.4			
Soignet, 1998 [76]	12	33.5	(9–75)			ATO 0.06–0.2 mg/kg/d				16.7	0		
Soignet, 2001 [29]	40			60.0		ATO 0.15 mg/kg/d				25.0	0		
Jin, 2006 [85]	30		(18–65)	60.0		ATO 10 mg qd				30.0	0	13.9 *	(5–25)
Shigeno, 2005 [80]; Ohnishi, 2002 [83]	34	47	(17–82)	64.7		ATO 0.15 mg/kg/d				23.5	0		
DiNardo, 2018 [13]	258	68	(18–89)	53.1	dose-escalation	ivosidenib 100 mg bid/300–1200 mg qd				11.2	0		
Stein, 2017 [91]; Fathi, 2018 [12]	239	70	(19–100)	57.3	dose-escalation	enasidenib 30–150 mg bid/50–650 mg qd				9.6		48	(10–340)
Tsai, 2008 [93]	27	69	(51–82)	70.4	dose-escalation	bexarotene 100–300 mg/m^2^, 400 mg/m^2^				7.4	0		
Tobita, 1997 [88]	24	49	(19–76)	54.2		tamibarotene 6 mg/m^2^/d bid				4.2	0	18	
Zhang, 2013 [84]	33	65	(60–79)							15.2	0		
					WBC ≤ 20 × 10^9/l^	ATO 0.16 mg/kg/d qd							
					WBC > 20 × 10^9/l^	ATO 0.08 mg/kg/d qd	daunorubicin 40 mg d1-3	cytarabine 50–100 mg d1-5					
Asou, 2007 [34]	283	48	(15–70)	55.8						21.2	0.7		
	85				WBC < 3 × 10^9/l^	ATRA 45 mg/m^2^/d tid							
	139				3 × 10^9/l^ ≤ WBC < 10 × 10^9/l^	ATRA 45 mg/m^2^/d tid	idarubicin 12 mg/m^2^/d qd d1-2	cytarabine 80 mg/m^2^/d d1-5					
	52				WBC > 10 × 10^9/l^	ATRA 45 mg/m^2^/d tid	idarubicin 12 mg/m^2^/d qd d1-3	cytarabine 100 mg/m^2^/d d1-5					
Asou, 2001 [75]	369	46	(15–85)	46.9						7.6	0.3		
	126			54.5	WBC < 3 × 10^9/l^	ATRA 45 mg/m^2^ tid							
	243			55.6	WBC ≥ 3 × 10^9/l^	ATRA 45 mg/m^2^ tid	daunorubicin 40 mg/m^2^/d qd 3d	enocitabine 200 mg/m^2^/d qd 5d					

**Abbreviations**: DS = Differentiation syndrome; TTO = Time to treatment onset, ATRA: All trans retinoic acid, ATO = arsenic trioxide, mg = milligrams, m^2^ = meters squared, d= day, tid, kg = kilograms, qd = once daily, bid = twice daily, tid = three times daily, qid = four times daily, qad = every other day; d5 = day 5 after start of treatment WBC = white blood cells. **Variables**: total = total number of patients in the clinical trial (arm); age = median age in years; allocatio*n* = criterion for the assignment of patients to the respective clinical trial arm; Primary treatment = describes drugs, doses, and frequencies of administration of the drugs administered for remission induction; %DS = proportion of patients who experienced DS in the respective clinical trial (arm); % deaths = proportion of patients who died as a consequence of DS; TTO = median (mean if marked with *) time to onset of DS after treatment start given in days. **Notes:** Information applicable to the whole clinical trial population is provided in the first row for each author. If individual trial arm data were available, the treatment-arm specific information is provided in the row’s underneath (one arm per row). Induction regimen description the dosing information is provided beneath the corresponding drug name. If a field is empty in the table, the value was not stated in the publication.

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
