# Peer review of "Incidence of Differentiation Syndrome Associated with Treatment Regimens in Acute Myeloid Leukemia: A Systematic Review of the Literature"

_jcm, 2020, doi:10.3390/jcm9103342_

Round 1

Reviewer 1 Report

This review is aimed to contextualise the occurrence of differentiation syndrome in clinical trials, with focus on the comparison of ivosidenib and enasidenib to existing ATRA and ATO therapies.

The authors did not observe difference between the ATRA- and ATO-based regimens used in the treatment of APL and they report that differentiation syndrome occurred less frequently during treatment with ivosidenib and enasidenib in non-M3 AML.

Minor Comments:

  • Why non-M3 AML patients are less prone to developing differentiation syndrome during their remission induction?
  • Why the authors decided to use only a single search engine (PubMed) for the scientific literature? Can they explain better?
  • Despite the dramatic impact of ATRA and ATO in the treatment of APL, deaths during induction from hemorrhage, infection, and differentiation syndrome continue to represent major causes for treatment failures in APL. The authors should comment this in the review.
  • Although the benefit of ATRA when combined with chemotherapy for consolidation has not yet been demonstrated in randomized studies,   trials by the GIMEMA and PETHEMA groups showed a statistically significant reduction in the relapse rate and a higher disease-free survival. The authors should report and comment these studies.
  • Tamibarotene was developed to overcome ATRA resistance and is currently being studied in combination with ATO in patients with relapsed disease. Any available data of the phase-2 American trial?

Author Response

Reviewer #1:

This review is aimed to contextualise the occurrence of differentiation syndrome in clinical trials, with focus on the comparison of ivosidenib and enasidenib to existing ATRA and ATO therapies.

The authors did not observe difference between the ATRA- and ATO-based regimens used in the treatment of APL and they report that differentiation syndrome occurred less frequently during treatment with ivosidenib and enasidenib in non-M3 AML.

Reviewer #1. Comment 1:

Why non-M3 AML patients are less prone to developing differentiation syndrome during their remission induction?

 Response:

The reviewer raises an important point, and we welcome the opportunity to expand upon this in our manuscript. Unfortunately, the factors that predict the development of DS are not well defined, neither in M3- nor in non-M3 AML. In the paper by Montesinos and Sanz (2011) it was identified that in M3-AML elevated levels of white blood cells, creatinine, lactate dehydrogenase, and peripheral blood blast count during treatment might be predictive of developing DS. While Leblebjian et al. (2013) also found the peripheral blood blast count to be predictive for DS, this was not confirmed by Elemam and Abdelmoety (2013), who found no significant indicators for the development of DS during APL treatment.

We further note that, as discussed in the paper [Discussion, page 13, lines 208 to 215] the incidence of DS in non-M3 patients treated with IDH-inhibitors is based solely on the published phase 1/2 clinical trials and might thus underestimate the real-world incidence of DS in these patients. Due to the novelty of DS in non-M3 AML patients’ signs and symptoms might not have been recognized and DS might have been underdiagnosed. Given the lack of data for these new IDH-inhibitor drugs, a definitive statement differences in propensity of developing DS between M3- and non-M3 AML patients cannot yet be made.

We have added the following to our introduction:

[page 2, lines 47-57] “It is estimated that up to a quarter of APL patients will experience DS during their treatment, yet evidence on the pathophysiology, epidemiology, predictive factors and prevention is till scarce.[1, 7, 8] Previously reported predictive factors of developing DS during treatment included elevated levels of white blood cells, creatinine, lactate dehydrogenase, and peripheral blood blast count [8] However, other studies did not or only partially corroborate these indicators for the development of DS during APL treatment.[9, 10] Overall prediction of development of DS is not well defined during APL treatment. While the overall prediction of development of DS is not well defined, data on DS in non-M3 patients treated with IDH-inhibitors are less understood and based on phase 1 and 2 trials. Thus, the signs and symptoms in this patient population is largely unknown. Further phase 3 clinical data and post-marketing studies, including pharmacovigilance data, could help enlighten this topic.”

Reviewer #1. Comment 2:

Why the authors decided to use only a single search engine (PubMed) for the scientific literature? Can they explain better?

Response

PubMed, which primary indexes medical literature and includes a search of the MEDLINE database, was used as it comprised more than 30 million citations. The single-engine search has been addressed in the limitations, which we have altered slightly for improved clarity as follows (altered first sentence in italics):

[page 14, lines 259-266] “We performed a single-engine search in PubMed, which primary indexes medical literature and comprised more than 30 million citations.[105] As such it is possible that we may have missed some articles that are not PubMed indexed. However, within PubMed we chose very broad search terms with no exclusion based on type of disease. This approach allowed investigation into an array of different drugs linked to DS, which was crucial considering that this systematic review was driven by recent unexpected findings of DS with new agents and outside of APL. Additionally, we conducted a thorough citation search of all relevant articles, which identified 25 additional articles, in order to minimize this risk.”

Reviewer #1. Comment 3:

Despite the dramatic impact of ATRA and ATO in the treatment of APL, deaths during induction from hemorrhage, infection, and differentiation syndrome continue to represent major causes for treatment failures in APL. The authors should comment this in the review.

Response:

We agree with the reviewer regarding the importance of the occurrence of hemorrhage and infection, in addition to differentiation syndrome, as important causes of treatment failure in APL. We have provided a further comment to this in the manuscript.

[page 1, lines 41-45] “Despite the dramatic impact of ATRA and ATO in the treatment of APL, major causes of treatment failures during induction therapy include fatalities resulting from haemorrhage and infection as general complications of chemotherapeutic therapy. Moreover, the adverse drug reaction “differentiation syndrome” is a serious and very specific complication that can result from ATRA treatment of APL.”

Reviewer #1. Comment 4:

Although the benefit of ATRA when combined with chemotherapy for consolidation has not yet been demonstrated in randomized studies,   trials by the GIMEMA and PETHEMA groups showed a statistically significant reduction in the relapse rate and a higher disease-free survival. The authors should report and comment these studies.

Response:

While we agree with the reviewer that the relapse rate and disease-free survival with these drugs is an important clinical endpoint, the focus of our review was on induction therapy as this is the only phase where differentiation syndrome occurs. Thus, a full review of the efficacy of the different drugs was not included. However, we believe this point is worth mentioning in our manuscript, we have therefore added the following sentence to our discussion:

[page 14, lines 278-283] “We also focused our review on the occurrence of DS, and therefore did not summarize information on disease-free survival or relapse rate. We note that the benefit of ATRA combination with conventional chemotherapy has not been demonstrated for consolidation in randomized studies, however, clinical trials have shown a statistically significant reduction in the relapse rate and longer disease-free survival.[32, 72]

Reviewer 1. Comment 5:

Tamibarotene was developed to overcome ATRA resistance and is currently being studied in combination with ATO in patients with relapsed disease. Any available data of the phase-2 American trial?

Response:

We thank the reviewer for raising this point. To the best of our knowledge, the results for the trial mentioned in our paper (NCT02807558) have not been reported (study completion expected in December 2021). Thus, we included the results from the phase 2 American trial (NCT00520208) by Sanford and colleagues (2015) on of tamibarotene in APL patients relapsed or refractory to treatment with ATRA and ATO.

We have added the following to the Discussion section of our paper to address this phase 2 study by Sanford et al.:

[page 13, lines 244-247] “Results of a multicenter phase 2 study on tamibarotene in adults with relapsed or refractory APL after treatment with ATRA and ATO (NCT00520208) found an overall response rate of 64%. However, the median overall survival was 9.5 months, and 7 of 9 patients relapsed after a median of 4.6 months.[104]”

Reviewer 2 Report

Gasparovic et al. reports the incidence of DS in patients with APL treated with various APL induction regimens between 15-20%. In addition, they also reviewed the incidence of DS in AML patients treated with IDH inhibitors, about 10%. DS is very well known complication of APL induction and IDH inhibitors. Authors should better describe the aim of this study, and their main point? As a clinician treating APL patients, I see limited value of this report in its current format.

The title is too long, and does not represent exactly what the paper is about

Figure 1 – difficult to follow, need to put arrows or lines to describe

Line 118: (Error! 119 Reference source not found) : use non-bold characters and explain in full sentence

Table 1. very busy table, has a lot of information , ok to put it in supplements, but authors should shorten it highlighting only clinically relevant items as a concise table

Author Response

Reviewer #2:

Gasparovic et al. reports the incidence of DS in patients with APL treated with various APL induction regimens between 15-20%. In addition, they also reviewed the incidence of DS in AML patients treated with IDH inhibitors, about 10%. DS is very well known complication of APL induction and IDH inhibitors. 

Reviewer #2. Comment 1:

Authors should better describe the aim of this study, and their main point? As a clinician treating APL patients, I see limited value of this report in its current format.

Response:

We thank the reviewer for directing our attention to a lack of clarity in the description of the aim of our study and welcome the opportunity to further clarify. As no comprehensive overview on the drugs associated with differentiation syndrome was performed from the scientific literature, we have aimed at closing this gap with the present systematic review. We therefore aimed to summarize the occurrence of differentiation syndrome in association with different treatment regimens in order to contextualise the occurrence of DS with older treatment options and also with the new IDH-inhibitors.

The focus on comparing different treatment regimens causing DS improves the understanding and highlights the significance of the sudden occurrence of DS in newly developed differentiating agents. As a consequence, we hope to raise awareness of DS in clinicians treating patients with novel differentiating agents and in researchers undertaking clinical studies with potential differentiating agents in development. This, we think, is necessary, considering that differentiating therapy is a promising and evolving therapeutic field [2]and considering the unspecific nature of the symptoms of DS which make it difficult to diagnose if occurring unexpectedly.

We added the following to the aim of the study in the Introduction section for more clarity:

[page 2, lines 80-83] “In particular we focus on the new IDH-inhibitor drugs as compared to existing differentiating therapies based on ATRA and ATO with the aim of contextualizing the occurrence of DS in these new agents. This is of particular interest given the recent and unexpected occurrence of DS observed with the IDH-inhibitors.”

Furthermore, we added the following to better address the clinical relevance of our research to the Conclusion:

[page 14, lines 300-303] “In light of the unexpected occurrence of DS outside of M3 AML and considering that differentiation therapy is a promising research area expected to contribute to future cancer treatments, we consider it crucial to raise awareness of DS in treating clinicians and in researchers with the aim of closing this knowledge gap.”

Reviewer #2. Comment 2:

The title is too long, and does not represent exactly what the paper is about

Response

We agree with the reviewer that the specificity of the title to the M3 subtype of acute myeloid leukaemia could lead to confusion on the content of the publication. Since the aim of this systematic review was to characterize the occurrence of DS irrespectively of the patient population and to capture all drugs DS is associated with, we have changed the title to the following:

“Treatment regimens associated with differentiation syndrome: A systematic review of the literature”

We believe that the revised title better captures the aim of the study while simultaneously being more concise.

Reviewer #2. Comment 3:

Figure 1 – difficult to follow, need to put arrows or lines to describe

Response

We apologize for the poor legibility of Figure 1, the article selection process flowchart, which seems to have stemmed from a compatibility problem with Word. The flowchart should of course have included lines and arrows already and we thank the author for pointing us to this formatting issue.

We have replaced the flowchart with a .png file and hope this resolves the issue. For convenience in reviewing, we have included the flowchart hereinafter, although no changes have been made to the content of the flowchart.

Reviewer #2. Comment 4:

Line 118: (Error! 119 Reference source not found) : use non-bold characters and explain in full sentence

Response

We thank the reviewer for pointing out this error in formatting. We have replaced the cross-reference to Figure 1 in line 119 with plain text. The content of the sentence has not been changed.

Reviewer #2. Comment 5:

Table 1. very busy table, has a lot of information , ok to put it in supplements, but authors should shorten it highlighting only clinically relevant items as a concise table

Response

We agree with the reviewer that Table 1 contains a lot of information, where we tried to reduce content without loss of major information. Unfortunately, limiting the numbers of rows is not possible without losing important information on treatment arms. However, we have aimed to reduce the number of columns in the table.

Reviewer 3 Report

In the paper “Differentiation syndrome associated with treatments used in the M3 subtype of acute myeloid leukaemia: A systematic review of the literature”, Gasparovic and colleagues performed a meta-analysis in order to analyze Differentiation  syndrome (DS) incidence in premyelocitic leukemia patients treated with pro-differentiation drugs. They compared these results with DS incidence in AML patients treated with other pro-differentiation drugs such as IDH-inhibitor drugs. Cumulative incidence of DS in all therapies was 17.7%, in trials with IDH-inhibitors was 10.4% while in trials with ATRA and/or ATO was 15.4%-20.6%.

The paper is well written but I completely disagree about the conception of the study and the usefulness of the results. In the paper, the authors compared DS incidence in two very different subtypes of leukemia (M3 and non-M3 acute leukemia) which are treated with very different drugs. ATRA/ATO are used in APL leukemia which If untreated, the median survival of is one-week while IDH inhibitors are used in “non-M3” leukemia IDH1/IDH2 mutated usually resistant/relapsed. I think there is no overlap between diseases, drugs or patients in this comparison, so the results are useless. Moreover, the authors did not try to find a biological, clinical, molecular, demographic link between these two different diseases. Results about DS incidence after ATRA/ATO treatment in acute promyelocytic leukemia (APL) are well known that are around 20-25% and this study doesn’t add any useful information to DS characterization (about symptoms, treatment regimen, age, survival, etc).

How these results could be useful for a clinician? Which is the message of this paper to the scientific community? Is Differentiation syndrome dangerous for an APL patient? Are there some marker that identify a greater risk of DS?

My comments are

  • The title is “Differentiation syndrome associated with treatments used in the “M3 subtype” of acute myeloid leukaemia: A systematic review of the literature”, but one of the major results of your paper is the lower DS incidence in trials with IDH-inhibitors and retinoic acide derivatives such as bexarotene. Unfortunately in the text you affirmed that “It is to be noted that the IDH-inhibitors, contrary to the ATRA- and ATO-based regimens, were used in the treatment of non-M3 AML, as was the retinoic acid derivative bexarotene.” There is no overlap between diseases, drugs or patients in your comparison. Why you compared these two different leukemias? Why in the title you talk only about M3 leukemias but in your results you analyzed also non-M3 patients?
  • I have some concerns about your results. In the paper you performed a risk of bias assessment and you found that the 82.1% of the trials analyzed showed high-risk of bias. In this context also your DS incidence analysis has high-risk of bias. Moreover, the number of ATRA/ATO and other treatments trials is unbalanced. In both IDH inhibitor and retinoic acid derivatives you used only 2 trials with high-risk of bias. The data are too poor to produce statistically robust results.
  • The paper using a big amount of data performed analyses only in DS incidence. How about remission rate, survival and symptoms related to DS? Did you try to estrapolate other informations about the characterization of DS?

Author Response

Reviewer #3:

In the paper “Differentiation syndrome associated with treatments used in the M3 subtype of acute myeloid leukaemia: A systematic review of the literature”, Gasparovic and colleagues performed a meta-analysis in order to analyze Differentiation  syndrome (DS) incidence in premyelocitic leukemia patients treated with pro-differentiation drugs. They compared these results with DS incidence in AML patients treated with other pro-differentiation drugs such as IDH-inhibitor drugs. Cumulative incidence of DS in all therapies was 17.7%, in trials with IDH-inhibitors was 10.4% while in trials with ATRA and/or ATO was 15.4%-20.6%.

The paper is well written but I completely disagree about the conception of the study and the usefulness of the results. In the paper, the authors compared DS incidence in two very different subtypes of leukemia (M3 and non-M3 acute leukemia) which are treated with very different drugs. ATRA/ATO are used in APL leukemia which If untreated, the median survival of is one-week while IDH inhibitors are used in “non-M3” leukemia IDH1/IDH2 mutated usually resistant/relapsed. I think there is no overlap between diseases, drugs or patients in this comparison, so the results are useless. Moreover, the authors did not try to find a biological, clinical, molecular, demographic link between these two different diseases. Results about DS incidence after ATRA/ATO treatment in acute promyelocytic leukemia (APL) are well known that are around 20-25% and this study doesn’t add any useful information to DS characterization (about symptoms, treatment regimen, age, survival, etc).

Response:

We appreciate and respect the opinion of the reviewer, however, would disagree that it is not worth assessing differentiation syndrome in both the treatment of ATRA/ATO and IDH-inhibitors.

As no comprehensive overview on the drugs associated with differentiation syndrome was performed from the scientific literature, we have aimed at closing this gap with the present systematic review. This paper constitutes the first systematic review and analysis of incidence of DS. Additional, given the recent and unexpected occurrence of DS in the novel IDH-inhibitors, we aimed to summarize the occurrence of differentiation syndrome in association with different treatment regimens in order to contextualise the occurrence of DS with older treatment options (ATRA/ATRO) and also with the new IDH-inhibitors.

The focus on comparing different treatment regimens causing DS improves the understanding and highlights the significance of the sudden occurrence of DS in newly developed differentiating agents. As a consequence, we hope to raise awareness of DS in clinicians treating patients with novel differentiating agents and in researchers undertaking clinical studies with potential differentiating agents in development. This, we think, is necessary, considering that differentiating therapy is a promising and evolving therapeutic field and considering the unspecific nature of the symptoms of DS which make it difficult to diagnose if occurring unexpectedly.

We added the following to the aim of the study in the Introduction and discussion sections for more clarity:

[page 2, lines 80-83] “In particular we focus on the new IDH-inhibitor drugs as compared to existing differentiating therapies based on ATRA and ATO with the aim of contextualizing the occurrence of DS in these new agents. This is of particular interest given the recent and unexpected occurrence of DS observed with the IDH-inhibitors.”

[page 14, lines 284-294] “Finally, while we acknowledge the limitation of comparing treatment regimens for M3 and non-M3 acute leukemia, we believe a review comparing different treatment regimens causing DS improves the understanding of this potentially fatal adverse drug reaction and highlights the significance in the newly developed IDH-inhibitors. We therefore hope to raise awareness of DS among treating clinicians and encourage researchers to conduct observational studies to provide real-world clinical evidence to further elucidate the development of DS in both M3 and non-M3 AML. Moreover, future research undertaking clinical development of potential differentiating agents should consider and assess the potential for DS. We believe this is necessary considering that differentiating therapy is a promising and evolving therapeutic field, [19] yet the unspecific nature of the symptoms of DS make it difficult to diagnose if occurring unexpectedly in non-traditional patient groups.”

How these results could be useful for a clinician? Which is the message of this paper to the scientific community? Is Differentiation syndrome dangerous for an APL patient? Are there some marker that identify a greater risk of DS?

Response

The reviewer raises an important point and we apologize that this was not clear in the original manuscript. As this is the first review of the IDH inhibitors, we are cautious with our conclusions. Nevertheless, we believe the primary message to clinicians is that differentiation syndrome is likely not uncommon in non-M3 type AML patients. However, with the limited phase 1 and phase 2 trial data, we believe stronger large clinical trial data, pharmacovigilance data, and real-world observational cohort data are required to better elucidate the risk of DS in this patient population.

Additionally, the delayed time to onset observed in one trials could indicate two possibilities – either the development of DS is later in these patients compared to M3-AML patients, or detection was delayed. In either situation, we would conclude that treating oncologists should be aware of this potentially fatal adverse drug reaction to ensure timely diagnosis and treatment to minimize the risk of severe outcome or death.

In order to clarify the main message of our publication, we have added the following sections to our discussion:

[page 13, lines 215-217] “Additionally, the delayed time-to-onset observed in the IDH-inhibitors indicates that either the development of DS is later compared to M3-AML patients, or detection was delayed.”

[page 14, lines 300-303] “In light of the unexpected occurrence of DS outside of M3 AML and considering that differentiation therapy is a promising research area expected to contribute to future cancer treatments, we consider it crucial to raise awareness of DS in treating clinicians and in researchers with the aim of closing this knowledge gap.”

Additionally, while the exploration of predictive factors for the development of DS was outside the scope of our study, we have added the following to clarify the current state of knowledge to the Introduction:

[page 2, lines 47 to 57] “It is estimated that up to a quarter of APL patients will experience DS during their treatment, yet evidence on the pathophysiology, epidemiology, predictive factors and prevention is till scarce.[1, 7, 8] Previously reported predictive factors of developing DS during treatment included elevated levels of white blood cells, creatinine, lactate dehydrogenase, and peripheral blood blast count [8] However, other studies did not or only partially corroborate these indicators for the development of DS during APL treatment.[9, 10] Overall prediction of development of DS is not well defined during APL treatment. While the overall prediction of development of DS is not well defined, data on DS in non-M3 patients treated with IDH-inhibitors are less understood and based on phase 1 and 2 trials. Thus, the signs and symptoms in this patient population is largely unknown. Further phase 3 clinical data and post-marketing studies, including pharmacovigilance data, could help enlighten this topic.”

Reviewer #3. Comment 1:

The title is “Differentiation syndrome associated with treatments used in the “M3 subtype” of acute myeloid leukaemia: A systematic review of the literature”, but one of the major results of your paper is the lower DS incidence in trials with IDH-inhibitors and retinoic acide derivatives such as bexarotene. Unfortunately in the text you affirmed that “It is to be noted that the IDH-inhibitors, contrary to the ATRA- and ATO-based regimens, were used in the treatment of non-M3 AML, as was the retinoic acid derivative bexarotene.” There is no overlap between diseases, drugs or patients in your comparison. Why you compared these two different leukemias? Why in the title you talk only about M3 leukemias but in your results you analyzed also non-M3 patients?

Response

We thank the reviewer for bringing up this important point. We agree with the reviewer that our choice of title had the potential to cause confusion considering that our study addressed both M3- and non-M3 AML. We have changed the title as follows:

“Treatment regimens associated with differentiation syndrome: A systematic review of the literature”

With the revised title, we hope to bring clarity to our approach to this systematic review. Clinical manifestations, diagnoses and treatments for AML and APL are not within the scope of this review, where we tried to focus on the adverse drug reaction DS. The aim of our study was to investigate all drugs which have been reported in clinical trials to cause DS. We therefore agree with the reviewer that the overlap is not in the disease, the drug or the patients, but in that these different patient populations, treated with different drugs, all have experience DS as a complication of their treatment for different forms of leukaemia.

As stated above to the general comments, we have added the following to the aim of the study in the Introduction section for more clarity:

[page 2, lines 80-83] “In particular we focus on the new IDH-inhibitor drugs as compared to existing differentiating therapies based on ATRA and ATO with the aim of contextualizing the occurrence of DS in these new agents. This is of particular interest given the recent and unexpected occurrence of DS observed with the IDH-inhibitors.”

Reviewer #3. Comment 2:

I have some concerns about your results. In the paper you performed a risk of bias assessment and you found that the 82.1% of the trials analyzed showed high-risk of bias. In this context also your DS incidence analysis has high-risk of bias. Moreover, the number of ATRA/ATO and other treatments trials is unbalanced. In both IDH inhibitor and retinoic acid derivatives you used only 2 trials with high-risk of bias. The data are too poor to produce statistically robust results.

Response

The reviewer is correct that the majority of studies were observed to have a high risk of bias, which is naturally of concern. However, as we noted in the results, this was primarily driven by a high risk of bias in the randomization domain due to the large inclusion of non-randomized trials. As random allocation is one criterion in the Risk of Bias scale, all (82.1%) non-randomized studies automatically fall into a high-risk category. This is shown in the paper in Figure 3 which we have provided below for convenience. We would argue that this does not automatically disqualify these studies, and is perhaps a limitation to the tool.

Thus, the high risk of bias in the randomization domain did not necessarily stem from poor conduct of the randomization process, but is rooted in the alternative approaches used for treatment allocation or the early stage of the clinical trial. We have elaborated in the Results and Discussion sections of our paper:

[page 10, lines 165 to 169] “The overall high risk of bias seen in most of the trials was primarily driven by concerns in the randomization process domain (Figure 3). It is to be noted that most of the included clinical trials were not randomized but instead used a risk- or age-based approach for assignment of the intervention or did not have a control arm, for example the included phase 1 and 2 trials of ivosidenib and enasidenib.”

[page 13, lines 248-253] “Due to the inclusion of many non-randomized clinical trials and the high risk of bias in the randomization process domain automatically resulting thereout, most of the clinical trials in this systematic review were concluded to have a high risk of bias. The high risk of bias noted does thus not necessarily disqualify these studies or their results but rather present a limitation to the assessment tool. For a better estimate of the real risk of bias, additional criteria could be included to evaluate the appropriateness of the treatment allocation process.”

Lastly, we would like to address the reviewer’s concern regarding the unequal distribution of clinical trials investigating different treatment regimens. With the aim of identifying all different treatment regimens in which DS has been reported to have occurred as a complication in clinical trials, our emphasis in the literature search and selection process lay on including all clinical trials which reported DS. While this has, as the reviewer correctly noticed, led to an imbalance of the number of clinical trials dedicated to each of the different treatment regimens, it has on the other hand allowed us to gain a comprehensive overview of different drugs causative of DS, despite the scarcity of clinical data on some of these drugs.

We would further like to note that the number of clinical trials retrieved on the different treatment regimens reflects the history of treatment recommendations for APL therapy, as is stated in the Discussion of our systematic review:

[page 12, lines 194-198] “In our review, we identified that the majority of studies and treatment arms were for the combination use of ATRA and chemotherapy. This was an expected finding, as it was the recommended first-line treatment regimen for all APL patients prior to the approval of ATO in combination with ATRA for the first-line treatment of low- and intermediate-risk APL in 2018.[2, 98]”

Reviewer #3. Comment 3:

The paper using a big amount of data performed analyses only in DS incidence. How about remission rate, survival and symptoms related to DS? Did you try to estrapolate other informations about the characterization of DS?

Response

We thank the reviewer for this comment, however, the primary aim of the manuscript was to focus on DS incidence as there were new reports on DS for newer substances such as IDH- and FLT3-inhibitors leading to signals raised by regulatory agencies. For these newer substances the incidence was estimated lower compared to older substances. However, data were obtained only from single clinical trials without systematic overview from various sources.

As reviewer 3 noted in comment 1, given the different AML types and thus different patient populations observed in the systematic review, we have decided on a cautious approach to the interpretation of our data and against extrapolation or speculation on DS characteristics for which the data was not given in the clinical trial literature.

Round 2

Reviewer 3 Report

In the revised version of the paper “Differentiation syndrome associated with treatments used in the M3 subtype of acute myeloid leukaemia: A systematic review of the literature”, Gasparovic and colleagues have changed several sections in the paper in order to reply my comments.

The study design is the same and all the revisions are sentences that try to clarify the paper aim.

My comments are:

  • The title is important because it explains the aim of the study. All my first review comments were based on that title and the aims depicted semantically from it.

I appreciated that the authors changed the title, because the paper did not talk about only M3-AML. Moreover, in the reply the authors affirm that “the primary aim of the manuscript was to focus on DS incidence as there were new reports on DS for newer substances such as IDH- and FLT3-inhibitors leading to signals raised by regulatory agencies”. Unfortunately, in the old title the main goal was the differentiation syndrome, thus I expected that in the review you examined several aspects of DS, not only the incidence. In the new title, I suggest to add the word “incidence” in order to describe better the real aim of the study.

  • My concerns about the comparison between ATRA/ATO and IDH-inhibitors still remain. DS in ATRA/ATO treatment has been studied from decade while clinical trials for IDH-inhibitors started 3 years ago. Montesinos criteria (Blood. 2009 Jan 22;113(4):775-83.) proposed 10 years ago diagnostic marks that identify DS in ATRA/ATO. How to apply these diagnostic criteria to other agents that may lead to clinical signs and symptoms of myeloid differentiation in AML is still unknown. For example, in an FDA study from Norsworthy (Clin Cancer Res. 2020 Aug 15;26(16):4280-4288.), the authors devised an algorithm, adapted from the Montesinos criteria in APL, to identify the incidence of DS in the combined 393 patients treated with either ivosidenib or enasidenib. After adjudication by the investigators, DS was identified in 19% of patients treated, considerably higher than the incidence of DS noted in previously published studies and very similar to ATRA/ATO studies. In this context, it seems that DS could be relatively common in patients treated with IDH inhibitors, likely underreported.

Moreover, ATRA/ATO complete remission (CR) rates reached 90–95%, while in IDH-inhibitors CR rates is around half. Maybe, the nonspecific symptoms of DS could be masked by features disease progression.

My concerns about statistical strength still remain. I hope that your results will be confirmed in the future studies.

Author Response

Response to Reviewer’s comments

Comment 1:

The title is important because it explains the aim of the study. All my first review comments were based on that title and the aims depicted semantically from it.

I appreciated that the authors changed the title, because the paper did not talk about only M3-AML. Moreover, in the reply the authors affirm that “the primary aim of the manuscript was to focus on DS incidence as there were new reports on DS for newer substances such as IDH- and FLT3-inhibitors leading to signals raised by regulatory agencies”. Unfortunately, in the old title the main goal was the differentiation syndrome, thus I expected that in the review you examined several aspects of DS, not only the incidence. In the new title, I suggest to add the word “incidence” in order to describe better the real aim of the study.

Response:          

We thank the reviewer for their suggestion to specify the focus on incidence in the title and agree that it better reflects the scope of our review. We have changed the title to the following:

Incidence of differentiation syndrome associated with treatment regimens in acute myeloid leukaemia: a systematic review of the literature

Comment 2:

My concerns about the comparison between ATRA/ATO and IDH-inhibitors still remain. DS in ATRA/ATO treatment has been studied from decade while clinical trials for IDH-inhibitors started 3 years ago. Montesinos criteria (Blood. 2009 Jan 22;113(4):775-83.) proposed 10 years ago diagnostic marks that identify DS in ATRA/ATO. How to apply these diagnostic criteria to other agents that may lead to clinical signs and symptoms of myeloid differentiation in AML is still unknown.

Response

We thank the reviewer for raising this important aspect that we thus far did not consider in the discussion of our findings. To address this point, we have added the following to the discussion:

[page 13, after line 209-216] “Lastly, due to the lack of diagnostic markers, the diagnosis of DS in ATRA- and ATO treatment relies on several unspecific symptoms. Whether these diagnostic criteria can be applied to detect DS in non-M3 AML patients reliably, or whether the application of the same criteria might lead to further underdiagnosis, is unclear. Moreover, we note that the complete remission rates were substantially higher in the ATRA/ATO patients than those treated with IDH-inhibitors. Thus, without clinical awareness and standardized diagnostic criteria of DS with IDH-inhibitors, it is possible that the symptoms of DS are incorrectly categorized as disease progression and subsequently under-recorded.”

Furthermore, to clarify the same point, we have added the following to the conclusion:

[page 14, lines 285-288] “While we found no difference between the ATRA- and ATO-based regimens used in APL treatment, DS occurred less frequently during treatment with IDH-inhibitors in non-M3 AML. However, this may be due to underreporting and lack of existing diagnostic criteria in this patient group.

Comment 3:

For example, in an FDA study from Norsworthy (Clin Cancer Res. 2020 Aug 15;26(16):4280-4288.), the authors devised an algorithm, adapted from the Montesinos criteria in APL, to identify the incidence of DS in the combined 393 patients treated with either ivosidenib or enasidenib. After adjudication by the investigators, DS was identified in 19% of patients treated, considerably higher than the incidence of DS noted in previously published studies and very similar to ATRA/ATO studies. In this context, it seems that DS could be relatively common in patients treated with IDH inhibitors, likely underreported.

Moreover, ATRA/ATO complete remission (CR) rates reached 90–95%, while in IDH-inhibitors CR rates is around half. Maybe, the nonspecific symptoms of DS could be masked by features disease progression.

Response:

We believe the reviewer has raised an important point regarding the recent findings of Norsworthy et al. We had included these findings in the discussion of the paper.

[page 13, lines 205-209] “A recent systematic analysis by the US FDA conducted an algorithmic analysis to improve the capture of DS cases in patients using IDH-inhibitors as many of the signs and symptoms of DS may not be initially recognized.[16] The authors concluded that DS was likely common with the IDH-inhibitors, and increased awareness of the early signs and symptoms is needed to decrease the potential for severe or fatal complications.”

However, to address the reviewer’s comments, as stated above in the answer to comment 2, we have added the following to the same paragraph of the discussion:

[page 13, after line 209-213] “Lastly, due to the lack of diagnostic markers, the diagnosis of DS in ATRA- and ATO treatment relies on several unspecific symptoms. Whether these diagnostic criteria can be applied to detect DS in non-M3 AML patients reliably, or whether the application of the same criteria might lead to further underdiagnosis, is unclear.”

To incorporate the important remark that disease progression might play a role in masking the nonspecific features of DS in non-M3 AML patients who achieve substantially lower CR rates, we have further added the following:

[page 13, lines 213-216] “Moreover, we note that the complete remission rates were substantially higher in the ATRA/ATO patients than those treated with IDH-inhibitors. Thus, without clinical awareness and standardized diagnostic criteria of DS with IDH-inhibitors, it is possible that the symptoms of DS are incorrectly categorized as disease progression and subsequently under-recorded.”

Comment 4:

My concerns about statistical strength still remain. I hope that your results will be confirmed in future studies.

Response:

We agree with the reviewer that their statistical concerns are valid given the relative lack of data in this area, especially so in the IDH-inhibitors due to their novelty. Nonetheless, we hope that with our review we manage to highlight the importance of DS awareness in new differentiating agents and the need for further research in this area to prevent future underdiagnosis of this potentially fatal complication. To contribute to this, our group will aim to add to this body of evidence with a pharmacovigilance analysis of global reports.